# DNAJA3 Interacts with PEDV S1 Protein and Inhibits Virus Replication by Affecting Virus Adsorption to Host Cells

**DOI:** 10.3390/v14112413

**Published:** 2022-10-31

**Authors:** Jingyou Zheng, Qin Gao, Jidong Xu, Xiaohan Xu, Ying Shan, Fushan Shi, Min Yue, Fang He, Weihuan Fang, Xiaoliang Li

**Affiliations:** 1Department of Veterinary Medicine, College of Animal Sciences, Zhejiang University, Hangzhou 310058, China; 2Zhejiang Provincial Key Lab of Preventive Veterinary Medicine, MOA Key Laboratory of Animal Virology, Center for Veterinary Sciences, Zhejiang University, Hangzhou 310058, China

**Keywords:** porcine epidemic diarrhea virus, DNAJA3, spike protein S1, viral adsorption

## Abstract

Porcine epidemic diarrhea virus (PEDV) infection causes huge economic losses to the pig industry worldwide. DNAJA3, a member of the Hsp40 family proteins, is known to play an important role in the replication of several viruses. However, it remains unknown if it interacts with PEDV. We found that DNAJA3 interacted with PEDV S1, initially with yeast two-hybrid screening and later with Co-IP, GST pull-down, and confocal imaging. Further experiments showed the functional relationship between DNAJA3 and PEDV in the infected IPEC-J2 cells. DNAJA3 overexpression significantly inhibited PEDV replication while its knockdown had the opposite effect, suggesting that it is a negative regulator of PEDV replication. In addition, DNAJA3 expression could be downregulated by PEDV infection possibly as the viral strategy to evade the suppressive role of DNAJA3. By gene silencing and overexpression, we were able to show that DNAJA3 inhibited PEDV adsorption to IPEC-J2 cells but did not affect virus invasion. In conclusion, our study provides clear evidence that DNAJA3 mediates PEDV adsorption to host cells and plays an antiviral role in IPEC-J2 cells.

## 1. Introduction

Porcine epidemic diarrhea (PED) caused by the PED virus (PEDV) was first reported in the United Kingdom in 1971 [1] and has been prevalent throughout the world since the 1990s [2,3]. Vaccines, either inactivated or attenuated, cannot provide sufficient protection against the currently circulating PEDV variant strains [4,5]. PEDV is still the main pathogen causing diarrhea in pigs [6,7,8]. PEDV belongs to the Alphacoronavirus genus with a genome size of about 28 kb encoding four structural proteins, namely spike (S), envelope (E), membrane (M), and nucleocapsid (N) proteins; 16 non-structural proteins (nsp1–nsp16); and the accessory protein ORF3 [9]. The S protein is responsible for viral entry into target cells [10] and plays a decisive role in tissue and cell tropism [11]. Variations in the S gene are often associated with significant changes in tropism and virulence and are a potential risk factor for cross-species transmission of the virus [12]. PEDV enters the target cells in two ways: direct entry upon fusion of the virions with the plasma membrane [13,14] and receptor-mediated endocytosis followed by the fusion of the viral membrane with an endosome [15,16]. The ectodomain of the S protein includes S1 and S2 subunits which are responsible for specific receptor binding and cell membrane fusion, respectively [17]. It has been demonstrated that PEDV S1 interacts with putative polypeptide receptors such as aminopeptidase N (APN), angiotensin-converting enzyme 2 (ACE2), and carbohydrate receptors such as heparin sulfate [18] and sialic acid [19,20]. PEDV S1 also plays a role in inducing cell apoptosis [21].

The DnaJ heat shock protein family (Hsp40) member A3 (DNAJA3), also known as Tid1, plays an important protective role in cells [22]. As a co-chaperone and regulator of heat shock protein 70 (Hsp70), DNAJA3 can regulate protein synthesis, folding, and membrane translocation by stimulating the ATPase activity of HSP70, and it participates in NF-κB signaling [23,24]. Knockout of Tid1 may result in massive cell death and embryonic death [25] while its overexpression could inhibit cell proliferation, migration, and invasion in vitro [26]. Human hTid1 inhibits apoptosis by interacting with adenomatous polyposis coli (APC) to antagonize the apoptotic function of the N-terminal region of the APC protein [27]. In addition, Hsp40 family proteins perform different functions in regulating the replication of different viruses [28]. DnaJB1 promotes influenza A virus replication by facilitating the nuclear import of viral ribonucleoproteins [29]. Hdj2 directly associates with the Japanese encephalitis virus NS5 to facilitate viral replication [30]. Conversely, Hsp40 protein can interact with the hepatitis B virus core protein to inhibit viral replication [31]. However, whether DNAJA3 affects PEDV replication remains uncharacterized.

In this study, a yeast two-hybrid assay identified host cell DNAJA3 as a specific binding partner of PEDV S1, which was further confirmed by co-immunoprecipitation (Co-IP), GST pull-down, and colocalization. The middle segment of S1 (aa378–479) was found to interact with DNAJA3. Subsequently, we have provided evidence that DNAJA3 mediated adsorption of PEDV to IPEC-J2 cells, thus affecting viral infectivity.

## 2. Materials and Methods

### 2.1. Cell Lines, Virus, and Antibodies

HEK293T, Vero E6, and IPEC-J2 cell lines were preserved in our laboratory. HEK293T and Vero E6 cells were cultured in Dulbecco’s Modified Eagle Medium (DMEM) (Gibco, Waltham, MA, USA) supplemented with 10% fetal bovine serum (FBS) (Yeasen, Shanghai, China) and antibiotics (100 U/mL penicillin, 100 µg/mL streptomycin, and 0.25 µg/mL amphozone) (Gibco, Waltham, MA, USA). IPEC-J2 cells were grown in a DMEM/nutrient mixture F-12 (DMEM/F12) (Gibco, Waltham, MA, USA) supplemented with 10% FBS and antibiotics. All cells were cultured at 37 °C and 5% CO_2_ atmosphere. The PEDV strain ZJ15XS0101 (GenBank accession No. KX550281.1) isolated from a clinically diseased pig was propagated and titrated in Vero E6 cells [32]. Mouse anti-PEDV-N monoclonal antibody (mAb), mouse anti-PEDV-S1 mAb, and rabbit anti-DNAJA3 polyclonal antibody (pAb) were prepared in our laboratory. Mouse anti-β-actin mAb was purchased from Beyotime (Shanghai, China), and the mAbs to GST, HA, or Flag tag from ABclonal (Wuhan, China).

### 2.2. Plasmid Construction

The S1 coding region was amplified with RT-PCR using PEDV RNA as a template and cloned into vectors (pGBKT7, pCMV-HA) with a ClonExpress II one-step cloning kit (Vazyme, Nanjing, China) to construct the recombinant plasmids pGBKT7-S1 and pCMV-S1-HA. The recombinant plasmids pCMV-S1T1 (amino acids(aa) 1 to 279), -S1T2 (aa 1 to 379), -S1T3 (aa 1 to 479), and -S1T4 (aa 1 to 649) with an HA tag were constructed in the same way. The DNAJA3 gene was amplified with RT-PCR using the full-genome RNA of IPEC-J2 cells as a template and cloned into vector pCMV-3×Flag using the homologous recombination kit. All recombinant plasmids were verified with DNA sequencing. The required primer sequences are shown in Table 1.

### 2.3. Yeast Two-Hybrid Assay

The yeast two-hybrid screen followed the Matchmaker Gold Yeast Two-Hybrid System User Manual (Clontech, Mountain View, CA, USA). The PEDV S1 gene fragment was cloned into pGBKT7 as the bait vector pGBKT7-S1 to express the fusion protein with the GAL4 DNA-binding domain. The cDNAs prepared from porcine enteroids were cloned into pGADT7 as the prey vector to express the fusion protein with the GAL4-activation domain (AD). The yeast strain Y2HGold was transformed with the pGBKT7-S1 bait using Yeastmaker™ Yeast Transformation System 2 (Clontech, Mountain View, CA, USA) and then mated with the Y187 strain transformed with the pGADT7-based porcine cDNA library (OE Biotech, Shanghai, China). Transformants were selected on plates with a defined medium lacking adenine, histidine, leucine, and tryptophan (SD/–Ade/–His/–Leu/–Trp) (quadruple dropout medium, QDO). The recovered colonies were collected for further screening on QDO plates containing 5-bromo-4-chloro-3-indolyl-α-d-galactopyranoside and aureobasidin A (QDO/X/A). Positive clones on the QDO/X/A plates were subjected to sequencing analysis and one-to-one verification that the bait and prey plasmids were co-transformed into the Y2HGold strain, resulting in the blue positive colonies on QDO/X/A when the bait and prey interacted.

### 2.4. Co-IP Assay

To examine the intracellular interaction between S1 and DNAJA3, HEK293T cells were transfected with the recombinant plasmids pCMV-S1-HA and pCMV-DNAJA3-Flag using Lipofectamine 2000 Reagents (Invitrogen, Carlsbad, CA, USA). The cells were lysed for Co-IP at 24 h post-transfection. The lysates were incubated with 2 mg of mouse anti-HA antibody, mouse anti-Flag antibody, or the same amount of mouse IgG as the control; rocked end-over-end overnight at 4 °C; and then mixed with Protein A+G Agarose (Beyotime, Shanghai, China) and rocked for 6–8 h. The agarose beads were washed with phosphate buffered saline (PBS) and boiled in a 1 × SDS sample buffer. The supernatant was obtained and detected with Western blotting using rabbit anti-HA or Flag mAb.

### 2.5. GST Pull-Down

The GST-tagged DNAJA3 and His-tagged PEDV S1 were expressed and purified using an *E. coli* expression system. At 100 µg each, both proteins were added into a 1.5-mL Eppendorf tube and the volume was made up to 1 mL with PBS to allow adequate incubation at 4 °C on a shaker. Then, 100 µL of BeyoGold™ GST-tag Purification Resin (Beyotime, Shanghai, China) was added to the protein mixture, and it continued to incubate at 4 °C for 3 h to allow its binding to DNAJA3-GST. The resin mixture was washed eight times with PBS to thoroughly remove unbound proteins. A volume of 50 μL of SDS-PAGE loading buffer (1×) was added to the resin, boiled for 5 min, and centrifuged at 12,000× *g* for 1 min. The supernatant sample was collected for subsequent detection by SDS-PAGE and Western blotting using mouse anti-PEDV-S1 mAb and mouse anti-GST mAb.

### 2.6. Immunofluorescence Assay (IFA)

An immunofluorescence assay was carried out to visualize the colocalization of DNAJA3 and S1 proteins as previously described [33]. Cells infected with PEDV (MOI = 1) for 24 h were fixed, permeabilized, and probed with mouse anti-PEDV-S1 mAb, mouse anti double-stranded RNA mAb, and a rabbit DNAJA3 antibody and then labeled with secondary antibodies, either Alexa 488- or Alexa 596-conjugated (ThermoFisher, Carlsbad, CA, USA). The nuclei were stained with DAPI (1 µg/mL). IPEC-J2 cells that were co-transfected with pCMV-DNAJA3-mCherry and pCMV-S1-eGFP for 24 h were stained with DAPI directly. Fluorescence images were captured with a laser confocal microscope IX81-FV1000 (Olympus, Tokyo, Japan).

### 2.7. Effect of PEDV Infection on DNAJA3 Expression

The IPEC-J2 were infected with different doses of PEDV (MOI = 0.5, 1 or 1.5) for 24 h or infected with PEDV at an MOI of 0.5 for different times (12, 24, or 36 h). The inoculated cells were incubated for 2 h at 37 °C and 5% CO_2_ to allow viral attachment. The infected cells were washed twice with DMEM and then cultured at 37 °C and 5% CO_2_ in DMEM containing 5 µg/mL of trypsin for the indicated time depending on the experimental requirements. The cell samples were collected followed by total RNA and protein extraction for analysis of DNAJA3 expression.

### 2.8. Effect of DNAJA3 on PEDV Replication

To examine if DNAJA3 had any effect on PEDV replication, we used gene knockdown and overexpression to modulate DNAJA3 expression. Small interfering RNAs (siRNA) targeting DNAJA3 listed in Table 1 were synthesized by GenePharma (Shanghai, China). The DNAJA3 siRNA was transferred into IPEC-J2 cells with GP-transfect-Mate provided by GenePharma to knockdown DNAJA3. Scramble RNA was used as a negative control. The overexpression of DNAJA3 was carried out by transfecting pCMV-DNAJA3-Flag into IPEC-J2 cells using Lipofectamine 2000 Reagents. The pCMV-Flag plasmid was used as a vector control. At 24 h post-transfection, the IPEC-J2 cells were infected with PEDV at an MOI of 1 for an additional 24 h. The cell samples were then harvested for analysis of DNAJA3 and N protein levels by Western blotting as well as the transcriptional levels of DNAJA3 and viral genome copies by quantitative reverse-transcription PCR (qRT-PCR) and cell supernatants for detection of viral titers.

### 2.9. Effects of DNAJA3 on PEDV Adsorption and Internalization

IPEC-J2 cells transfected with pCMV-DNAJA3-Flag or DNAJA3 siRNA for 24 h were pre-chilled at 4 °C for 30 min. Then, the cell monolayers were inoculated with PEDV (MOI = 5) and incubated at 4 °C for another 1 h to allow the virus’s adsorption but not internalization [34]. After washing with ice-cold PBS three times to remove the unbound virus, the total RNA was extracted to detect viral S1 gene transcription by RT-qPCR to evaluate the effect of DNAJA3 on PEDV adsorption. In another set of the cells, the virus-containing medium was replaced with fresh DMEM containing 5 µg/mL of trypsin, and the cells were transferred to a 37 °C incubator with 5% CO_2_ to allow viral internalization for 1 h. The cells were carefully washed once with a citrate buffer (pH 3.0) and then washed three times with PBS to remove the non-internalized virus. Finally, the total RNA was extracted for RT-qPCR as above to evaluate the effect of DNAJA3 on PEDV internalization.

In order to explore the effect of DNAJA3 protein on PEDV infection, the PEDV with an estimated MOI of 1 was mixed with prokaryotic DNAJA3 protein of different concentrations (5, 25, and 50 μg/mL) with the same levels of GST as the controls, and the mixture was incubated at 4 °C for 2 h. The mixtures were then added into IPEC-J2 cells in 24-well plates. After being infected in DMEM with 4 μg/mL of trypsin at 37 °C for 24 h, the cells were treated as above for the analysis of PEDV N transcription and N protein level using the aforementioned methods.

### 2.10. RT-qPCR

The total RNA was extracted from the cells using the RNA Extraction Kit (Bioteke, Beijing, China). The RNA concentration and purity was determined using NanoDrop (Thermo Scientific, Waltham, MA, USA). For determination of viral RNA copies, the RNA samples were directly used as a template for quantitative PCR detection with HiScript One Step qRT-PCR Probe Kit (Vazyme, Nanjing, China). To detect the transcription level of DNAJA3 and GAPDH, the RNA samples were reverse transcribed into cDNA with HiScript II Q RT SuperMix for qPCR (+gDNA wiper) (Vazyme, Nanjing, China) according to the manufacturer’s protocol. RT-qPCR analysis was performed with the ChamQ Universal SYBR qPCR Master Mix (Vazyme, Nanjing, China) on a Mx3005P Real-Time PCR system (Agilent, Santa Clara, CA, USA). The GAPDH transcription level was used to normalize the data, and the relative DNAJA3 expression was analyzed using the cycle threshold (2−ΔΔCt) method.

### 2.11. SDS-PAGE and Western Blotting

The protein samples were separated by 12% SDS-PAGE gels and blotted onto the polyvinylidene difluoride (PVDF) membranes (Millipore, Burlington, MA, USA) for probing with different primary and secondary antibodies using the protocol as described above [32]. The target protein bands were visualized with FDbio-Dura ECL Kit (FD, Dalian, China) on the Gel 3100 chemiluminescent imaging system (Sagecreation, Beijing, China). Relative protein levels were estimated by densitometric analysis using the Image J software.

### 2.12. Viral Titration

Vero-E6 cells seeded in 96-well plates were cultured at 37 °C and 5% CO_2_ for 12~16 h. The virus-containing supernatant was diluted in a gradient 10-fold dilution with DMEM medium containing 5 μg/mL trypsin and inoculated into Vero cells at 100 μL/well. Eight replicate wells were set for each viral gradient dilution. After 3 days of incubation in the same conditions as above, the wells with a viral growth cytopathic effect were observed directly under the microscope and the remaining wells were visualized by probing with an anti-PEDV N mAb and secondary antibodies conjugated with Alexa Fluor 488. The viral titer was calculated as the 50% tissue culture infective dose (TCID50/mL) using the Reed-Muench method.

### 2.13. Statistical Analysis

All data was expressed as means ± standard deviations (SD) of three independent experiments. Statistical significance was tested with the Student’s *t*-test, and *p* values less than 0.05 were considered statistically significant.

## 3. Results

### 3.1. Host Protein DNAJA3 Interacted with PEDV S1

To analyze the interaction between PEDV S1 and host proteins, we screened the porcine enteroid cDNA yeast library using PEDV S1 as bait. A total of 15 proteins were identified as putative binding partners of S1 (Table 2). Co-transformation of the prey plasmid together with the bait plasmid (BD-S1) into the yeast Y2HGold strain was retested on QDO/X/A plates to eliminate some false positive clones (Figure 1A). DNAJA3 was chosen for further study due to its function in virus replication in early studies as shown below. The two-way Co-IP assay showed the interaction between S1 and DNAJA3 because the co-precipitated HA-tagged S1 protein or Flag-tagged DNAJA3 protein could be detected (Figure 1B,C). As shown in Figure 1D, there was obvious colocalization of DNAJA3 with S1 in the cytoplasm of IPEC-J2 cells. The colocation of DNAJA3 with viral dsRNA intermediate was also apparent in PEDV-infected IPEC-J2 cells (Figure 1E). In order to confirm the interaction, purified DNAJA3-GST and PEDV S1 proteins were used for the GST pull-down assay. DNAJA3 did interact directly with S1 (Figure 1F). The above results confirmed the interaction between DNAJA3 and PEDV S1 in vitro and in the cultured cells expressing both proteins or infected with PEDV.

To identify the regions of S1 that are responsible for the specific interaction with DNAJA3, a series of deletion mutants of S1 were constructed (Figure 2A). A Co-IP assay showed that S1 fragments 1–479 (T3) and 1–649 (T4) were able to interact with DNAJA3, similar to the S1 full-length, but those from 1–279 (T1) or 1–379 (T2) were not, suggesting that the middle fragment from the second half of S1 between aa379 and aa479 is required for interaction with DNAJA3 (Figure 2B).

### 3.2. PEDV Infection Inhibited Expression of DNAJA3

To examine the relationship between PEDV and DNAJA3, IPEC-J2 cells were infected with PEDV at an MOI of 0.5, 1, and 1.5 for 24 h or at 0.5 MOI for 12, 24, and 36 h to examine changes in DNAJA3 expression. DNAJA3 expression in the PEDV-infected cells decreased dose-dependently at both mRNA and protein level and showed a slight reduction at 0.5 MOI, but they showed marked reduction when increasing the MOI to 1.5 as compared to the mock-infected control (Figure 3A–C). PEDV infection at 0.5 MOI did not significantly inhibit the transcription of DNAJA3 mRNA until 24 h post-infection (hpi) and thereafter. At the protein level, reduced DNAJA3 expression was apparent only at 36 hpi (Figure 3D,E). Collectively, these results indicate that PEDV inhibition of DNAJA3 expression is associated with viral load.

### 3.3. DNAJA3 Repressed PEDV Replication

Previous studies have shown that DNAJA3 is involved in the replication of several viruses [35,36,37]. We wondered if DNAJA3 affects PEDV replication. The IPEC-J2 cells transfected with pCMV-DNAJA3-Flag were further infected with PEDV at 1.0 MOI. Increased DNAJA3 transcription was apparent (Figure 4A). As shown in Figure 4B, the viral RNA copies in DNAJA3-overexpressed cells were significantly lower than those in the cells transfected with the control vector, so there was a marked reduction of the viral titer in the culture supernatant (Figure 4C) and viral N protein in the cell lysates (Figure 4D). To further confirm the antiviral role of DNAJA3, gene silencing was also performed. As shown in Figure 5A,B, the knockdown efficiency of 3 siRNAs targeting on different sites of DNAJA3 was effective. DNAJA3-3 was selected for subsequent experiments (Figure 5C–G). DNAJA3 knockdown facilitated PEDV replication in IPEC-J2 cells, as evidenced by viral RNA copies, virus titer, and N protein expression (Figure 5D–F,H). Taken together, the above data demonstrate that DNAJA3 negatively regulates PEDV proliferation.

### 3.4. DNAJA3 Interfered Adsorption of PEDV to IPEC-J2 Cells

The life cycle of PEDV includes adsorption, endocytosis, replication, and virion release [38]. To examine if DNAJA3 affects PEDV infection at the adsorption or internalization stage, we pre-treated the IPEC-J2 cells with DNAJA3 silencing or overexpressing vectors which were then infected with a high infection dose of PEDV (MOI = 5) for adsorption at 4 °C for 1 h or shifted to 37 °C for 1 h to estimate virus entry. Silencing or overexpression was relatively effective (Figure 6A–D). The mRNA level of adsorbed virions was significantly increased in DNAJA3-konckdown cells compared to the control (Figure 6E), while no significant difference was detected in the mRNA level of internalized virions (Figure 6F). DNAJA3 overexpression significantly reduced the mRNA transcripts of the attached virus to the cells (Figure 6G). However, there was no apparent change in the mRNA transcripts of the internalized virus (Figure 6H). These findings suggest that DNAJA3 inhibits PEDV adsorption but not the internalization in IPEC-J2 cells.

### 3.5. Recombinant DNAJA3 Protein Suppressed PEDV Infection

Given that viral adsorption occurs on the host cell membrane, we presumed that part of the DNAJA3 could be present on the cell membrane. Confocal imaging suggests that DNAJA3 was localized on the cell membrane stained with the dye DiI (Figure 7A). We explored the effect of exogenous DNAJA3 protein on virus adsorption. We found that viral RNA and N protein abundance decreased significantly with an increasing amount of DNAJA3 used during pre-incubation with PEDV compared to the control cells treated with GST (Figure 7B–D), suggesting that exogenous DNAJA3 interfered with PEDV adsorption, thereby reducing the level of its subsequent replication in the cells.

## 4. Discussion

Similar to most coronaviruses, PEDV has a coronoid process formed with a trimer spike (S) protein [17]. Its binding with the receptor(s) is the initial event for the virus to invade cells, and it is also adept in using various components of host cells to promote replication and host pathogenesis [38]. Here, we show that DNAJA3, a member of the Hsp40 family, interacts with PEDV S1 and inhibits viral adsorption to the IPEC-J2 cells while PEDV infection could downregulate its expression.

The S1 protein was selected in this study as bait to screen host proteins in the porcine enteroid cDNA library because of its roles in binding to the host cells [38] and in interacting with the viral accessory protein ORF3, which is involved in PEDV replication [39]. Of the 15 host proteins showing putative binding with S1 from the cDNA library screening, 4 proteins clearly interacted with PEDV S1 through verification with paired yeast hybridization (Appendix A). DNAJA3 protein attracted our attention since Hsp40 is involved in the replication of several viruses, such as human immunodeficiency virus [40,41,42], influenza A virus [29,43], Herpes simplex virus type 1 [44], etc. Zhang et al. demonstrated that DNAJA3 interacted with the VP1 of the food-and-mouth virus and inhibited viral replication by inducing lysosomal degradation of VP1 and weakening its antagonistic effect in the interferon β signaling pathway [37]. HSP40 has been reported to interact with the 3′ end of RNA of the murine hepatitis virus to possibly promote its stability [45]. However, there is a paucity of information on the role of Hsp40 family members in other coronaviruses. Through Co-IP and confocal imaging, we revealed that S1 did interact with DNAJA3, and the S1 region of aa379–479 was responsible for the binding.

Further experiments showed a functional relationship between DNAJA3 and PEDV in the infected IPEC-J2 cells. DNAJA3 overexpression significantly inhibited PEDV replication, while DNAJA3 knockdown had the opposite effect, suggesting that DNAJA3 is a negative regulator of PEDV replication. However, PEDV could repress DNAJA3 expression, seemingly in an attempt to evade the suppressive role of DNAJA3 to favor its replication. Xu et al. have demonstrated that the PEDV N protein interacts with Sp1 and interferes with its binding to the promoter region of HDAC1, thereby inhibiting its transcriptional activity [45]. How PEDV inhibits the expression of DNAJA3 requires further exploration. Because S1 is involved in viral invasion, most likely by interacting with yet-to-be identified receptor(s), we hypothesized that DNAJA3 might function on the cell surface by affecting the virus’s entry, either adsorption or internalization, during the viral infection. The short-time infection of the IPEC-J2 cells with high MOI at 4 °C allowed the stepwise analysis of viral attachment and invasion [34,46]. This permitted us to examine the roles of DNAJA3 in these processes by manipulating its expression levels, either by gene-silencing or overexpression. We found that DNAJA3 inhibited PEDV adsorption to IPEC-J2 cells but did not affect viral invasion (Figure 6). This implies that the binding of the S1 protein to the host cell receptor(s) might be compromised upon its interaction with DNAJA3. This could be supported by the facts that the pretreatment of PEDV with recombinant DNAJA3 inhibited viral replication, likely as a result of interference with an S1-mediated viral attachment upon the binding of DNAJA3 to the spike protein on the viral particle, and that DNAJA3 is present on the cell surface as shown by its colocalization with the membrane labelling dye Dil (Figure 7), although this molecular chaperone is mostly located in the cytoplasmic and mitochondrial compartments [47].

This is the first report providing evidence that DNAJA3 acts on the cell surface to interfere with viral adsorption and infectivity while there is a limited number of studies on its involvement in viral replication and infectivity inside the cells as described above. However, we could not rule out other possibilities that this protein functions in the cytoplasmic compartment, considering that its human counterpart hTid-1 participates in several protein–protein interactions in the cells which mediate different processes, such as proteasomal degradation, autophagy, etc. [48]. The hTid-1 represses the activity of NF-κB through physical and functional interactions with the IKK complex and IκB and modulates cell growth and death [49]. The hTid-1 isoforms are reported to interact with Jak2 and Hsp70/Hsc70 and modulate IFN-γ-mediated transcriptional activity, and the interaction between Hsp70/Hsc70 and hTid-1 is reduced after IFN-γ treatment [50]. Recent studies have indicated that such cellular events and signaling are involved during PEDV infection [38,51,52]. Further research is required to determine if and how DNAJA3 functions in regulating PDEV replication inside the cells.

In summary, we provide clear evidence that the direct interaction between DNAJA3 and S1 protein may inhibit adsorption of PEDV to the host cells while PEDV infection could downregulate DNAJA3 expression. These findings expand our understanding of PEDV–host interactions.

## Figures and Tables

**Figure 1 viruses-14-02413-f001:**
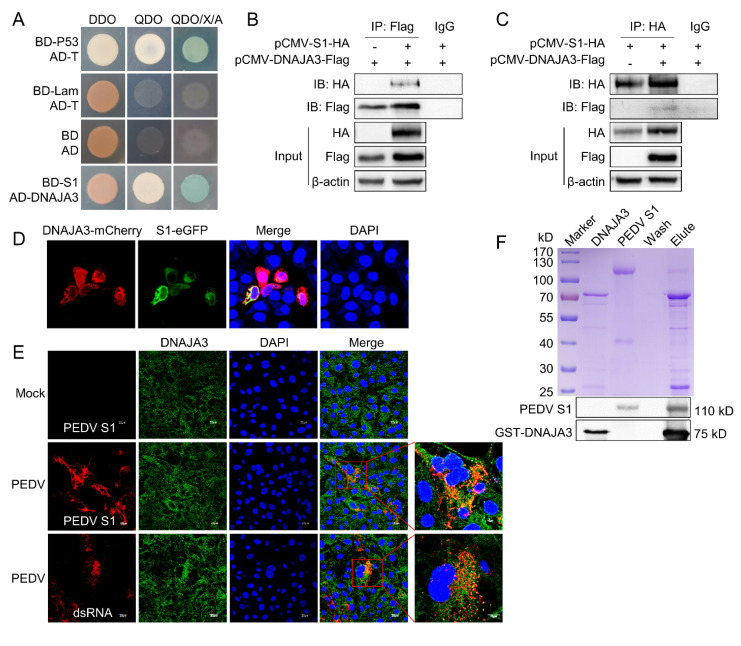
DNAJA3 protein interacted with PEDV S1. (**A**): Yeast strain Y2HGold was co-transformed with bait plasmid pGBKT7-S1 (BD-S1) and prey plasmid pGADT7-DNAJA3 (AD-DNAJA3). The blue bacterial lawn growing on the QDO/X/A plates represents positive interactions. Co-transformation of pGBKT7-53 (BD-p53) and pGADT7-T (AD-T) served as positive control, pGBKT7-Lam (BD-Lam) and pGADT7-T, as negative control, and pGBKT7 (BD) and pGADT7 (AD), as blank. DDO, SD/−Leu/−Trp; QDO, SD/–Ade/–His/–Leu/–Trp; QDO/X/A, SD/–Ade/–His/–Leu/–Trp/X-a-Gal/AbA. (**B**,**C**): HEK293T cells were transfected individually or co-transfected with plasmids expressing S1-HA and DNAJA3-Flag for two-way Co-IP. Cell lysates were immunoprecipitated with mouse anti-Flag or rabbit anti-HA mAb, followed by immunoblotting with rabbit anti-Flag pAb or mouse anti-HA mAb to reveal S1 and DNAJA3, respectively. (**D**): Confocal images showing colocalization of DNAJA3 and S1 in the cells transfected with pCMV-DNAJA3-mCherry and pCMV-S1-eGFP. (**E**): Colocalization of endogenous DNAJA3 and S1 in PEDV-infected IPEC-J2 cells. (**F**): GST-pull-down: GST-DNAJA3 and PEDV S1 protein were mixed and incubated overnight, and then bound with GSH-purification resin. The eluates were detected by Coomassie blue staining (top) and Western blotting (bottom) using mouse anti-GST and anti-S1 mAb.

**Figure 2 viruses-14-02413-f002:**
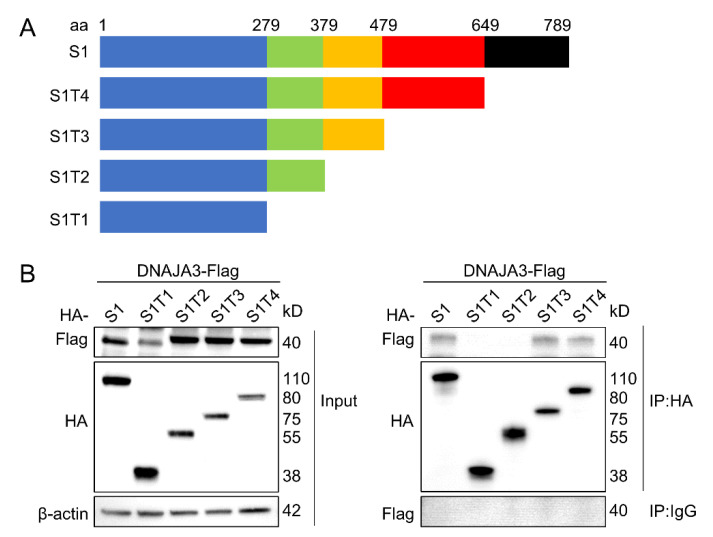
The middle fragment of PEDV S1 between aa379 and aa479 was the region to interact with DNAJA3. (**A**): Schematic representation of the full-length S1 protein and its truncated proteins. (**B**): Co-IP detection of the interaction of DNAJA3 with S1 protein region in the IPEC-J2 cells co-transfected with pCMV-DNAJA3-Flag and plasmids expressing S1 full-length or its truncated proteins.

**Figure 3 viruses-14-02413-f003:**
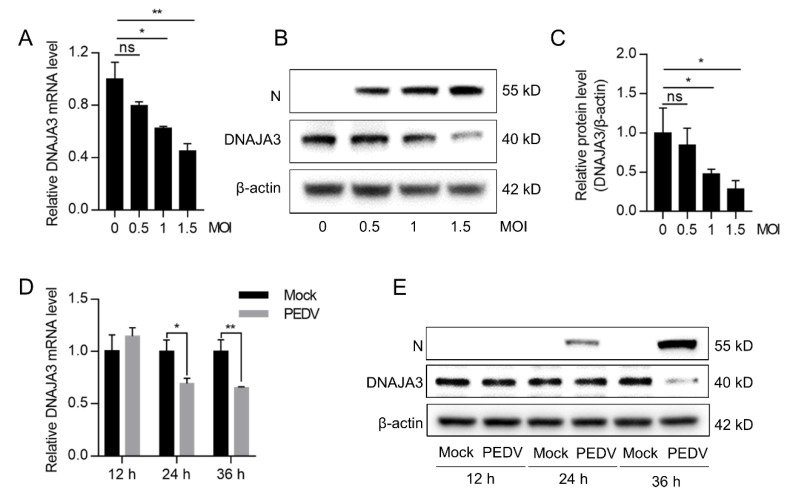
PEDV infection downregulated DNAJA3 expression in a dose and time-dependent manner. IPEC-J2 cells were infected with different MOI of PEDV. (**A**,**B**): At 24 h, relative mRNA level (**A**) and protein level (**B**) of DNAJA3 were detected by qPCR and Western blotting. (**C**): Ratio of DNAJA3 to β-actin of panel B by densitometric analysis. D&E: IPEC-J2 cells were infected with PEDV (0.5 MOI), and the mRNA level (**D**) and protein level (**E**) of DNAJA3 at 12, 24 and 36 h. All results are presented as the mean ± SD of data from three independent experiments. *, *p* < 0.05; **, *p* < 0.01.

**Figure 4 viruses-14-02413-f004:**
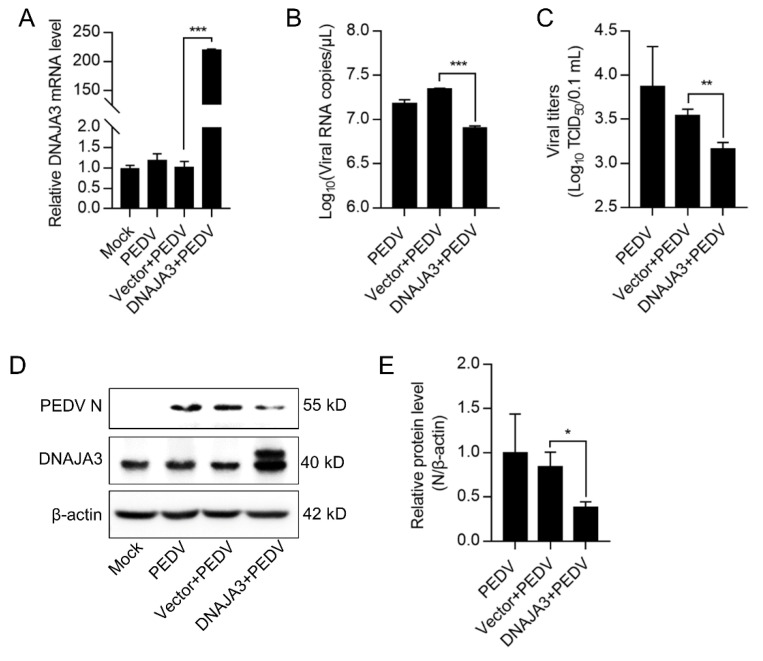
Transient expression of DNAJA3 inhibited PEDV replication. pCMV-DNAJA3 (DNAJA3) or pCMV-Flag (Vector) were transfected into IPEC-J2 cells for 24 h, respectively. Then the cells were infected with PEDV (1.0 MOI) for 24 h. (**A**): Relative DNAJA3 mRNA. (**B**): Viral genome copies. (**C**): PEDV titers (TCID50) in the supernatant. (**D**): Protein levels of DNAJA3 and PEDV-N by Western blotting. (**E**): Ratios of N protein to β-actin of panel D. The results were shown as means ± SD of three independent experiments; *, *p* < 0.05; **, *p* < 0.01; ***, *p* < 0.001.

**Figure 5 viruses-14-02413-f005:**
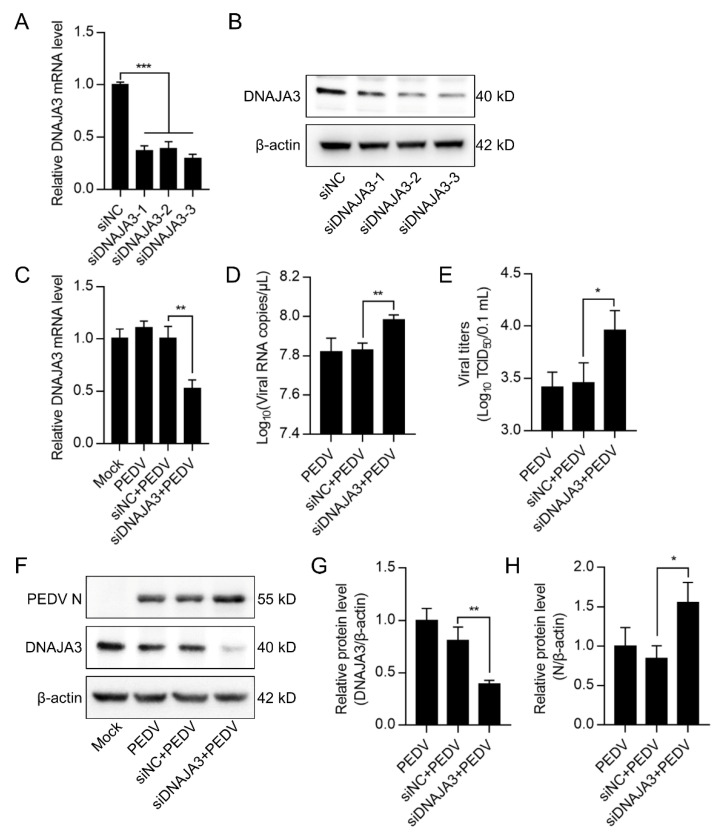
Knockdown of DNAJA3 facilitated PEDV replication. (**A**,**B**): IPEC-J2 cells were transfected with three interfering RNAs targeting DNAJA3 or with control vector (siNC) for 48 h. Downregulation of DNAJA3 by qPCR (**A**) and Western blotting (**B**). (**C**–**F**): IPEC-J2 cells were transfected with siDNAJA3-3 or siNC for 24 h and then infected with PEDV (MOI = 1) for additional 24 h for analysis of relative DNAJA3 transcription (**C**), viral RNA copy numbers (**D**), viral titers (TCID50) (**E**), and levels of PEDV N and DNAJA3 (**F**) in the infected cells. (**G**,**H**): Ratios of DNAJA3 (**G**) or N protein (**H**) to β-actin of panel F. All results are presented as the mean ± SD of data from three independent experiments. *, *p* < 0.05; **, *p* < 0.01; ***, *p* < 0.001.

**Figure 6 viruses-14-02413-f006:**
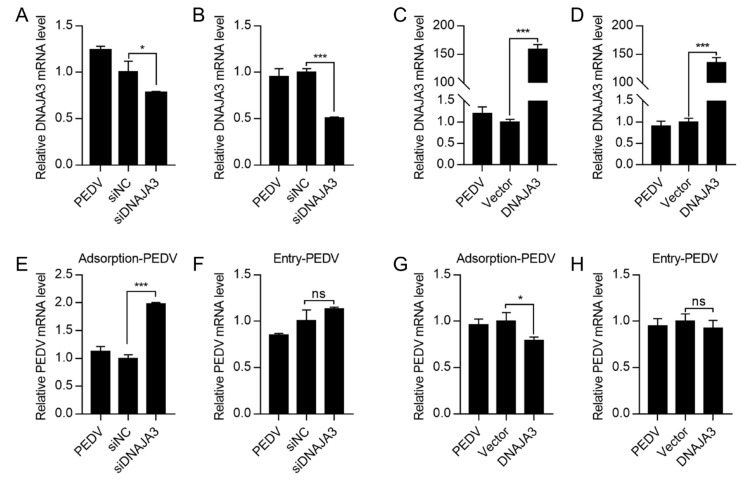
Effects of DNAJA3 on adsorption or internalization of PEDV. (**A**–**D**): IPEC-J2 cells transfected with gene silencing vector siDNAJA3 (with siNC as control) or with expression vector pCMV-DNAJA3 (DNAJA3) (with pCMV-Flag as Vector control) for 24 h and the transfected cells were then inoculated with PEDV (5.0 MOI) for 1 h at 4 °C to allow virus attachment without internalization. The cells in wells for adsorption were then washed and used to extract total RNA to detect mRNA levels of DNAJA3 (**A**,**C**). The cells in the remaining wells were cultured in fresh serum-free DMEM and subsequently shifted to 37 °C to allow virus internalization for 1 h. The cells were washed to remove non-internalized virions and used to total RNA of infected cells to detect mRNA levels of DNAJA3 (**B**–**D**). (**E**,**F**): Effects of DNAJA3 knockdown on PEDV adsorption and internalization shown as viral RNA copies. (**G**,**H**): Effects of DNAJA3 overexpression on PEDV adsorption and internalization shown as viral RNA copies. For all RT-qPCR analysis, the transcription levels of control vectors (siNC or Vector) were set to 100%. The results are presented as the mean ± SD of data from three independent experiments. ns, *p >* 0.05; *, *p* < 0.05; ***, *p* < 0.001.

**Figure 7 viruses-14-02413-f007:**
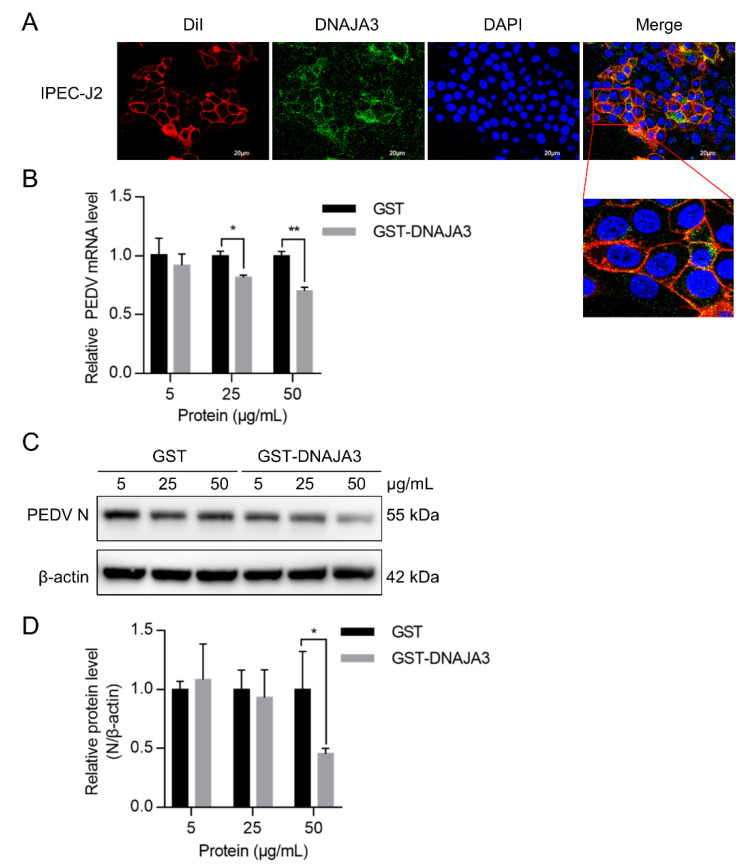
Pretreatment of PEDV with DNAJA3 protein inhibited its infection. (**A**): IPEC-J2 cells were analyzed to determine colocalization of fluorescent dye DiI that stains cell membranes (red) with DNAJA3 (green) with DAPI-nuclear staining (blue). (**B**,**C**): Different levels of GST or DNAJA3-GST protein (5, 25 and 50 μg) were mixed with PEDV (MOI = 1) and pre-incubated at 4 °C for 2 h. The IPEC-J2 cells were inoculated with the mixtures and incubated at 37 °C for additional 2 h to allow virus attachment. The culture supernatants were then replaced with DMEM containing 4 μg/mL trypsin and the plates continued to culture for 22 h. PEDV replication was shown as mRNA (**B**) and N protein (**C**) levels. (**D**): Ratios of N protein to β-actin of panel C. The results are presented as the mean ± SD from three independent experiments. *, *p* < 0.05; **, *p* < 0.01.

**Table 1 viruses-14-02413-t001:** Primers and DNAJA3 silencing sequence used in this study.

Primer	Orientation	Sequence (5′ to 3′)
Real-time PCR		
GAPDH	F	CACTGAGGACCAGGTTGTGTCCTGTGAC
	R	TCCACCACCCTGTTGCTGTAGCCAAATTC
DNAJA3	F	TCCGAGTTCCCAGGAGACTGA
	R	GTTTCCAGTGGACCGTTTTCCA
PEDV-N	F	CGGAACAGGACCTCACGCC
	R	ACAATCTCAACTACGCTGGGAAG
PEDV-S1	F	CGGTTTGTTGGATGCTGTC
	R	AATAAAGAATACGCTGAATGGC
siRNA		
siNC	F	UUCUCCGAACGUGUCACGUTT
	R	ACGUGACACGUUCGGAGAATT
siDNAJA3	F	GGUUGAGUGGGAAAGGCAUTT
	R	AUGCCUUUCCCACUCAACCTT
Plasimid construction		
pGBKT7-S1	F	ATGGAGGCCGAATTCATGAAGTCTTTAACCTACTTCTGGTTGT
	R	CAGGTCGACGGATCCTTAACTAAAGTTGGTGGGAATACTGATA
pGEX-4T-1-linearized	F	CCGGGTCGACTCGAGCG
	R	ACGCGGAACCAGATCCG
4T-1-DNAJA3	F	GATCTGGTTCCGCGTATGGCGGCGCGGTGC
	R	CTCGAGTCGACCCGGGTTTCCAGTGGACCGTTTTCCAG
pCMV-3*Flag-linearized	F	GGATCCCGGGCTGACTACAA
	R	AAGCTTAATTCTGACGGTTCAC
DNAJA3	F	GTCAGAATTAAGCTTATGGCGGCGCGGTGC
	R	GTCAGCCCGGGATCCGTTTCCAGTGGACCGTTTTCCAG
pCMV-HA	F	TACCCATACGATGTTCCAGATTACGCTTAAGCGGCC
	R	CTCGAGGCAGATCTCGGTCGACCGAATTC
S1-HA	F	GAGATCTGCCTCGAGATGAAGTCTTTAACCTACTTCTGGTTGT
	R	AACATCGTATGGGTAACTAAAGTTGGTGGGAATACTGA
S1T1-HA	R	AACATCGTATGGGTACAAAGGTTGGTTGGAAACCACC
S1T2-HA	R	AACATCGTATGGGTAAGTATCCACTTTAAGAAAACAATAATAGGGTAC
S1T3-HA	R	AACATCGTATGGGTAGTCAAAAGCAACCTGAGAACACTTG
S1T4-HA	R	AACATCGTATGGGTAACACACATCCAGAGTCATAAAAGAAACGTC

The underlined sequence is the homologous sequence corresponding to the two ends of the linearized cloning vector.

**Table 2 viruses-14-02413-t002:** Host proteins that interact with PEDV S1 protein as determined by yeast two-hybrid analyses.

Protein Name	Gene ID	No. of Clones
claudin-15 (CLDN15)	100302018	3
ribosomal protein L7a (RPL7A)	110255317	1
DnaJ heat shock protein family (Hsp40) member A3 (DNAJA3)	100624182	8
eukaryotic translation elongation factor 1 alpha 1 (eEF1A1)	574059	2
ER membrane protein complex subunit 10 (EMC10)	100526111	1
macrophage migration inhibitory factor (MIF)	397412	2
glutathione S-transferase A2 (GSTA2)	396850	1
double-headed protease inhibitor, submandibular gland-like	100739218	1
zinc finger and BTB domain containing 7A (ZBTB7A)	100625057	1
haloacid dehalogenase like hydrolase domain containing 3 (HDHD3)	106508518	1
lectin galactoside-binding soluble 3 (LGALS3)	100038033	1
SNARE-associated protein Snapin (Snapin)	100154273	1
NADH:ubiquinone oxidoreductase subunit A8 (NDUFA8)	100154443	2
glyceraldehyde-3-phosphate dehydrogenase (GAPDH)	396823	1
ribosomal protein S6 (RPS6)	100038023	1

## Data Availability

Not applicable.

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
