# Peer review of "DNAJA3 Interacts with PEDV S1 Protein and Inhibits Virus Replication by Affecting Virus Adsorption to Host Cells"

_viruses, 2022, doi:10.3390/v14112413_

Round 1
Reviewer 1 Report
This manuscript describes the host protein, DNAJA3, in regulating host innate immunity-PEDV infection interactions, which belongs to DanJ heat shock protein family (Hsp40). By performing yeast two-hybrid assay, Co-IP, and GST pull-down in vitro, the authors found for the first time that DNAJA3 interacts with PEDV S protein, which is responsible for virus entry into the target cells. Furthermore, the authors confirmed that DNAJA3 was significantly downregulated by PEDV infection in IPEC-J2 cells. This evidence drove the authors to study the functions of DNAJA3 in PEDV infection. By performing overexpression and gene knockdown, DNAJA3 was confirmed as a negative regulator in PEDV replication, probably by affecting PEDV adsorption to IPEC-J2 cells rather than PEDV invasion. Overall, DNAJA3 is indicated as a crucial host effector for anti-PEDV innate immune responses; the firm interaction of PEDV S protein with host DNAJA3 protein might contribute a potential mechanism for PEDV evading from host innate immune responses. My major concern is why the authors focused on DNAJA3 to study the host-virus interaction among 15 proteins identified as potential binding partners of PEDV S1 protein (as is shown in Table 2 and Line 228). Is DNAJA3 the only one that reported to function in virus replication? It might be more innovative to study the other unreported proteins in PEDV infection.
This study is well-executed and the data are consistent between experiments. The conclusion drawn by the authors is confirmed by multiple experimental methods. However, there are still some minor issues that needs to be improved in this manuscript.
Minor issues:
1. Line 30-31, “… was first reported…” should be “… was firstly reported…”
2. Line 129, the DNAJA3 and PEDV S1 proteins were all GST-tagged? This sentence is inaccurate and should be rewritten.
3. Line 311, “DNAJA3 negatively regulate PEDV proliferation” shoule be “DNAJA3 negatively regulates PEDV proliferation”.
4. Line 335 and elsewhere, “adsorbed virus” should be “adsorbed virions”.
5. In the Discussion section, do other Hsp40 family members function in anti-PEDV/other coronaviruses innate immune responses?
Reviewer 2 Report
In this article, Jingyou Zheng et al showed that DNAJA3, a member of Hsp40 family interacts with PEDV S1, and significantly inhibited PEDV replication. Furthermore, DNAJA3 could inhibit PEDV adsorption to IPEC-J2 cells. These data are detailed and strong, but some revisions still needed.
Major comments:
1. How did PEDV infection influence the mRNA level of DNAJA3? These authors should clarify and design experiments to confirm the hypothesis. Whether PEDV infection inhibits the promoter activity of DNAJA3? Or does PECV infection increase the degradation of DNAJA3 protein? Or both exist.
2. For PEDV infection inhibiting DNAJA3 mRNA and protein level, I am wondering which protein of PEDV performs the inhibitory function.
3. It’s hard to say that DNAJA3 inhibited PEDV replication, because as shown in the manuscript, DNAJA3 inhibited adsorption of PEDV.
4. In figure 7, it should be that recombinant DNAJA3 protein suppressed PEDV infection, but not replication.
Reviewer 3 Report
The article by Zheng et.al shows how DNAJA3, a member of Hsp40 family proteins interact with PED virus and inhibit viral replication. Here through yeast two-hybrid system, Co-IP, pulldown assay and confocal imaging the authors have shown that DNAJA3 interacted with PEDV.
Further using IPEC-J2 cells they have found functional relationship between DNAJ3 and PEDV. PEDV infection inhibited the expression of DNAJ3, whereas DNAJ3 repressed PEDV replication. By overexpressing and silencing they have found out that DNAJ# inhibited PEDV adsorption to cells but didn’t affect internalization. The article is well written. Just a minor suggestion please mention the plasmid deletion detail(amino acid deleted for SIT1 to S1T4) in material methods (plasmid construction)
Round 2
Reviewer 2 Report
All of my questions has been answered, this new revised version is much straightforward and easy to read and comprehend. It is my pleasure to give the green light for this publication.